# Transport evidence of asymmetric spin–orbit coupling in few-layer superconducting $1T_d$-MoTe$_2$

Jian Cui[1,11], Peiling Li[1,2,11], Jiadong Zhou[3,11], Wen-Yu He [4,11], Xiangwei Huang[1,2], Jian Yi[5], Jie Fan[1], Zhongqing Ji[1], Xiunian Jing[1,6], Fanming Qu[1], Zhi Gang Cheng [1], Changli Yang[1,6], Li Lu[1,6], Kazu Suenaga[7], Junwei Liu [4], Kam Tuen Law[4], Junhao Lin [8,9], Zheng Liu [3] & Guangtong Liu [1,10]

Two-dimensional transition metal dichalcogenides $MX_2$ ($M =$ W, Mo, Nb, and $X =$ Te, Se, S) with strong spin–orbit coupling possess plenty of novel physics including superconductivity. Due to the Ising spin–orbit coupling, monolayer NbSe$_2$ and gated MoS$_2$ of $2H$ structure can realize the Ising superconductivity, which manifests itself with in-plane upper critical field far exceeding Pauli paramagnetic limit. Surprisingly, we find that a few-layer $1T_d$ structure MoTe$_2$ also exhibits an in-plane upper critical field which goes beyond the Pauli paramagnetic limit. Importantly, the in-plane upper critical field shows an emergent two-fold symmetry which is different from the isotropic in-plane upper critical field in $2H$ transition metal dichalcogenides. We show that this is a result of an asymmetric spin–orbit coupling in $1T_d$ transition metal dichalcogenides. Our work provides transport evidence of a new type of asymmetric spin–orbit coupling in transition metal dichalcogenides which may give rise to novel super-conducting and spin transport properties.

[1] Beijing National Laboratory of Condensed Matter Physics, Institute of Physics, Chinese Academy of Sciences, 100190 Beijing, China. [2] University of Chinese Academy of Sciences, 100049 Beijing, China. [3] School of Materials Science and Engineering, Nanyang Technological University, Singapore 639798, Singapore. [4] Department of Physics, Hong Kong University of Science and Technology, Clear Water Bay, Hong Kong, China. [5] Ningbo Institute of Industrial Technology, Chinese Academy of Sciences, 315201 Ningbo, China. [6] Collaborative Innovation Center of Quantum Matter, 100871 Beijing, China. [7] National Institute of Advanced Industrial Science and Technology (AIST), Tsukuba 305-8565, Japan. [8] Department of Physics, Southern University of Science and Technology, 518055 Shenzhen, China. [9] Shenzhen Key Laboratory of Quantum Science and Engineering, 518055 Shenzhen, China. [10] Songshan Lake Materials Laboratory, Dongguan 523808 Guangdong, China. [11] These authors contributed equally: Jian Cui, Peiling Li, Jiadong Zhou, Wen-Yu He. Correspondence and requests for materials should be addressed to J.L. (email: lin.junhao.stem@gmail.com) or to Z.L. (email: z.liu@ntu.edu.sg) or to G.L. (email: gtliu@iphy.ac.cn)

In conventional Bardeen–Cooper–Schrieffer (BCS) singlet superconductor, the external magnetic field becomes detrimental to the superconductivity state through orbital depairing effect and Pauli paramagnetism[1]. In the two-dimensional (2D) atomically thin superconductor, the orbital effect is suppressed and Pauli paramagnetism plays the dominant role when an in-plane magnetic field is applied. The effect of in-plane magnetic field on the 2D superconductor is recently studied in the superconducting $2H$-type transition metal dichalcogenides (TMDCs), including gated $MoS_2$[2,3], 2D $NbSe_2$[4,5], and monolayer $TaS_2$[6]. Interestingly, the in-plane upper critical field $\left( H_{c2,\parallel} \right)$ is observed to be strongly enhanced beyond the Pauli paramagnetic limit[7,8]. The large enhancement of $H_{c2,\parallel}$ in the $2H$ TMDCs originates from the strong Ising spin–orbit coupling (SOC) due to the breaking of an in-plane mirror symmetry and the presence of the out-of-plane mirror symmetry in the crystal structure. As electron spins are pinned to the out-of-plane directions, this phenomenon is named Ising superconductivity. Ising superconductivity[2–4] with its promising applications in equal spin Andreev reflections[9], proximity phenomenon[10], engineering Majorana fermions[9,11,12], and topological superconductivity[13,14] has sparked intense research interest in condensed matter physics. So far, the study of SOC effect on superconductivity in TMDCs is limited to the Ising SOC.

In this work, we systematically study the superconducting few-layer $1T_d$-$MoTe_2$. A few-layer $1T_d$-$MoTe_2$, unlike its $2H$ structure counterparts, breaks both the in-plane mirror symmetry and out-of-plane mirror symmetry. We show that the resulting SOC is asymmetric in the three spatial directions. By combining the low-temperature transport measurements and self-consistent mean-field calculations, we demonstrate that the in-plane upper critical field in the superconducting few-layer $1T_d$-$MoTe_2$ exceeds the Pauli limit in the whole in-plane directions. Importantly, we theoretically predicted and experimentally verified that the in-plane upper critical field shows an emergent two-fold symmetry due to the new type of anisotropic SOC. From the experimental data, we further estimated that the SOC strength is on the orders of tens of meV, which is also consistent with the results of our first-principle calculations. Our work gives clear evidence that anisotropic SOC plays an important role in determining the properties of superconductivity in $MoTe_2$.

## Results

**Growth of few-layer $MoTe_2$ crystals.** In our experiment, the high crystalline few-layer $MoTe_2$ crystals were produced by molten-salt assisted chemical vapor deposition (CVD) method[15,16] (see details in the Methods section and Supplementary Fig. 1). The optical images of the as-synthesized $MoTe_2$ layers with different thicknesses are shown in Fig. 1a. Similar to our previous results[15], the mono- and few-layer $MoTe_2$ can have a size up to 100 μm with a rectangular shape. Figure 1b shows the Raman spectra of the as-synthesized $MoTe_2$ with different layers, where we ascribe the Ag modes at 127, 161, and 267 $cm^{-1}$ to $1T_d$-$MoTe_2$, which is further supported by the following scanning transmission electron microscopy (STEM) measurements. Note that the Ag mode located at 267 $cm^{-1}$ shows a blueshift with increasing sample thickness, which is similar to the Raman shift in other 2D materials such as $MoS_2$[17] and $WS_2$[18].

**Structural characterization.** The atomic structure of few-layer $MoTe_2$ is further characterized by annular dark-field (ADF) STEM imaging. Figure 1c shows the atom-resolved STEM image of few-layer $MoTe_2$ in a large scale (see Supplementary Fig. 2a, b for the chemical purity verified by energy-dispersive X-ray spectra). The $1T_d$ phase and $1T'$ phase share the same in-plane

crystal structure (Supplementary Fig. 2c), but the two structures have different stacking. The $1T'$ crystallizes in monoclinic shape and keeps the global inversion center, while the $1T_d$ phase has the vertical stacking and belongs to the non-centrosymmetric space group $Pmn2_1$, as shown by the atomic models in Fig. 1c. Therefore, the projection of the scattering potential is different in these two phases and can be distinguished by their STEM images. At room temperature, bulk $MoTe_2$ usually crystalized in the monoclinic $1T'$ phase. By comparing the simulated images both in $1T_d$ and $1T'$ phase shown in Fig. 1c, we unambiguously found that the few-layer $MoTe_2$ is in the $1T_d$ phase rather than the bulk $1T'$ phase at room temperature. This is different from the previous reports[19,20] where the $1T_d$ phase only occurs at temperature below 200 K. Additionally, the temperature dependence of Raman spectra and sheet resistance shown in Supplementary Fig. 3 also confirm the few-layer $MoTe_2$ is in the $1T_d$ phase, that is, no phase transition is observed on lowering down the temperature. Therefore, the $1T_d$ phase could be the intrinsic feature of the CVD-grown few-layer $MoTe_2$, which is presumably caused by the reduced thickness of $MoTe_2$[21].

**Transport properties of few-layer $1T_d$-$MoTe_2$.** Figure 1d shows the temperature dependence of the normalized four-terminal sheet resistance ($R/R_{300 K}$), measured at zero magnetic field, for $MoTe_2$ films with thickness from 2 to 30 nm (see Supplementary Fig. 4 for raw data). At high temperatures, all samples measured show a metallic behavior with $dR/dT > 0$, indicating that the phonon scattering dominates the transport. As the temperature is further lowered, the samples enter a disorder-limited transport regime prior to the eventual superconducting state. The residual resistance ratio, $RRR = R_{300 K}/R_n$ with $R_{300 K}$ the room temperature sheet resistance and $R_n$ the normal state sheet resistance right above the superconducting transition, which varies from 1.15 of the 2.0-nm-thick to 2.33 of the 30-nm-thick $MoTe_2$ crystals (Supplementary Table 1).

At low temperatures, superconductivity is observed for all samples. To examine the thickness-dependent superconductivity, in the inset of Fig. 1d we show the temperature dependence of the reduced resistance, $r = R/R_n = R/R_{5 K}$, in a low-temperature regime ($T \leq 5.5$ K) for samples with different thickness. Empirically, critical transition temperatures for the superconductivity, $T_{c,r}$, can be extracted from the $R$ vs. $T$ curve. This is realized by picking up the points firstly encountered with the predefined reduced resistance $r$ from the normal state into the superconducting state. Such transition temperatures, extracted at typical values $r = 0$, 0.5, and 0.9, are listed in Supplementary Table 1 for our samples with different thickness. It is found that, $T_{c,0}$ increase from 0.35 K to 3.16 K with increasing sample thickness from 2 nm to 30 nm, and the $T_{c,0}$ values of our samples are surprisingly higher than $T_c \sim 0.1$ K as reported in stoichiometric bulk $MoTe_2$[20]. In bulk $MoTe_2$, Te-vacancy-enhanced superconductivity has been previously reported[16] with the highest $T_{c,0} \sim 1.3$ K, which is still much lower than $T_{c,0} = 3.16$ K observed in our 30-nm-thick $MoTe_2$ crystals. In addition, a significant broadening on superconducting transition are observed for 2-nm-thick device, which can be attributed to the enhanced thermal fluctuations in two dimensions[1,22]; similar behaviors have been observed in few-layer $Mo_2C$[23] and $NbSe_2$[4,24] superconductors reported recently.

The few-layer $MoTe_2$ crystals provide an ideal platform to study their transport properties in the 2D limit. To investigate the dimensionality of the superconductivity in few-layer $MoTe_2$, we firstly studied the temperature dependence of the upper critical magnetic field $\mu_0 H_{c2}$, which is defined as the magnetic field corresponding to a predefined reduced resistance $r = R/R_n = 0.5$.

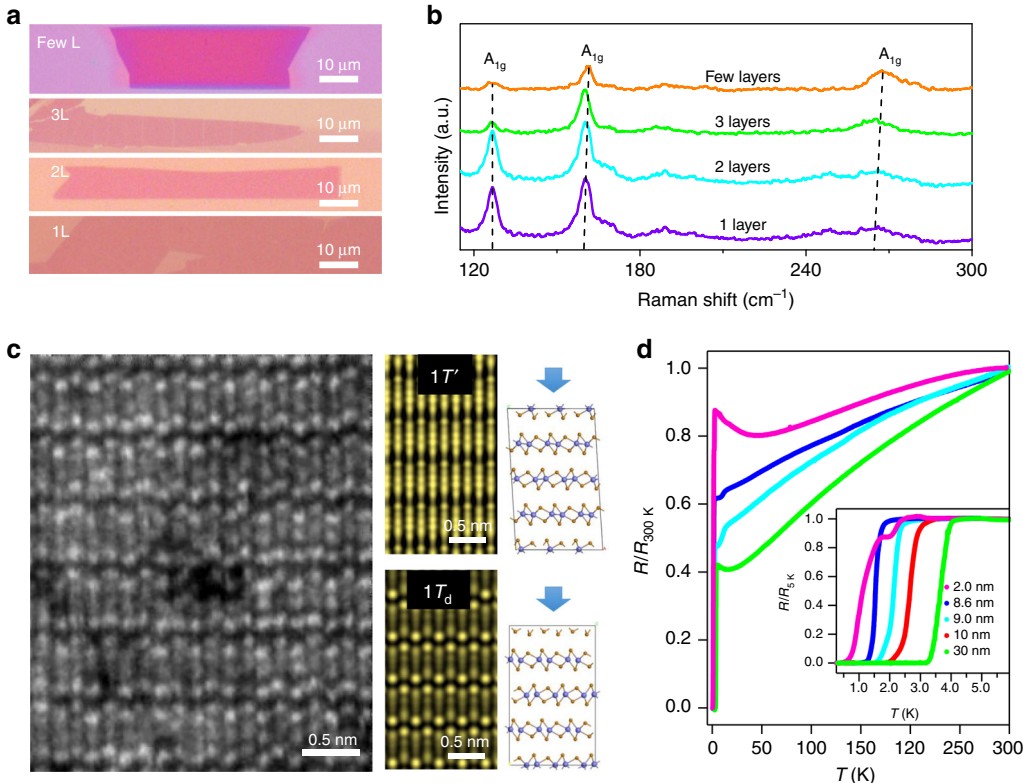

**Fig. 1** Structural and transport characterization of the as-synthesized $1T_d$-MoTe$_2$ samples. **a** Optical images of monolayer (1L), bilayer (2L), trilayer (3L), and few layers of as-synthesized MoTe$_2$. The size of the monolayer sample can reach up to 100 μm. The scale bar is 10 μm. **b** Raman spectra of the few-layer MoTe$_2$ samples. Raman peaks were observed at 127, 161, and 267 cm$^{-1}$, corresponding to the Ag modes of $1T_d$-MoTe$_2$. **c** Atomic resolution scanning transmission electron microscopy (STEM) image of few-layer $1T_d$-MoTe$_2$. Simulated STEM images of few-layer MoTe$_2$ in $1T'$ and $1T_d$ stacking viewed along [0 0 1] zone axis are shown next to the experimental image, respectively. Compared with the simulation, the stacking of the few-layer MoTe$_2$ is confirmed to be the $1T_d$ phase. The scale bar is 0.5 nm. **d** Superconductivity in few-layer $1T_d$-MoTe$_2$. The inset shows the temperature dependence of the reduced four-terminal resistance ($R/R_{5\,K}$) in the range from 0.3 to 4.5 K, for MoTe$_2$ devices with the thickness ranging from 2 to 10 nm

Due to the high reactivity of oxygen and water vapor, few-layer, especially monolayer MoTe$_2$, samples deteriorate easily in ambient conditions. The following data were mainly collected on samples with the thickness ranging from 2.7 to 9 nm. Figure 2a, b shows the superconducting resistive transitions of a 8.6-nm-thick MoTe$_2$ device with the magnetic field perpendicular and parallel to the sample surface, respectively, measured at fixed temperatures. In both cases, one can see that the superconducting transition shifts gradually to lower magnetic fields with the increase of temperature. The temperature-dependent upper critical fields in directions parallel and perpendicular to the sample surface, denoted by $\mu_0 H_{c2,\parallel}$ and $\mu_0 H_{c2,\perp}$, respectively, are plotted in Fig. 2c. We found that the superconductivity was more susceptible to perpendicular magnetic fields than to parallel magnetic fields, and a large ratio of $H_{c2,\parallel}/H_{c2,\perp} \approx 7$ is obtained in the 8.6-nm-thick sample, indicating a strong magnetic anisotropy. This is true for all samples, and the ratio reaches up to 26 for 2.7-nm-thick sample (see Supplementary Fig. 5 for more samples). A linear temperature dependence was observed for $H_{c2,\perp}$, which can be well fitted by the phenomenological 2D Ginzburg–Landau (GL) theory[1],

$$H_{c2,\perp}(T) = \frac{\phi_0}{2\pi\xi_{GL}^2}\left(1 - \frac{T}{T_{c,0}}\right), \tag{1}$$

where $\xi_{GL}$ is the zero temperature GL in-plane coherence length and $\phi_0$ is the magnetic flux quantum, as shown by the blue dashed line in Fig. 2c that gives $\xi_{GL} = 20.79$ nm (see Supplementary Table 1 for more data on samples with different thickness).

Compared with the sample thickness of 8.6 nm as measured by AFM (Supplementary Fig. 5f), the coherence length $\xi_{GL} = 20.79$ nm is approximately two times of the thickness, indicating that the depairing effect from the orbital magnetic field is strongly suppressed. As a result, the in-plane magnetic field-induced paramagnetism determines $H_{c2,\parallel}(T)$.

The 2D behavior of the superconducting few-layer MoTe$_2$ is further confirmed by the experiments with tilted magnetic field. Figure 2d shows the magnetic field dependence of the sheet resistance $R$ under different $\theta$ at 0.3 K, where $\theta$ is the tilted angle between the normal of the sample plane and the direction of the applied magnetic field (the inset of Fig. 2e). Clearly, the superconducting transition shifts to higher field with the external magnetic field rotating from perpendicular $\theta = 0^\circ$ to parallel $\theta = 90^\circ$ (see Supplementary Fig. 6 for more data on different MoTe$_2$ samples). The upper critical field $\mu_0 H_{c2}$ was extracted from Fig. 2d and plotted in Fig. 2e as a function of the tilted angle $\theta$. In Fig. 2e, a cusp-like peak is clearly observed at $\theta = 90^\circ$, where the external magnetic field is aligned in parallel to the sample surface, which is apparently sharper for thinner sample (Supplementary Fig. 6). Curves fitted with the 2D Tinkham model[1] and 3D anisotropic GL model show the data consistence with both models for $\theta < 85^\circ$ and $\theta > 95^\circ$, whereas for $85^\circ < \theta < 95^\circ$, the cusp-shaped dependence can only be explained with the 2D Tinkham model as shown in Fig. 2f. It shows that our superconducting few-layer MoTe$_2$ manifests the 2D nature of the superconductivity. For a 2D superconductor with $d_{sc} \ll \xi_{GL}$, the $V$-$I$ dependence as a function of temperatures is measured and shown in Supplementary Fig. 6. The Berezinskii–Kosterlitz–Thouless

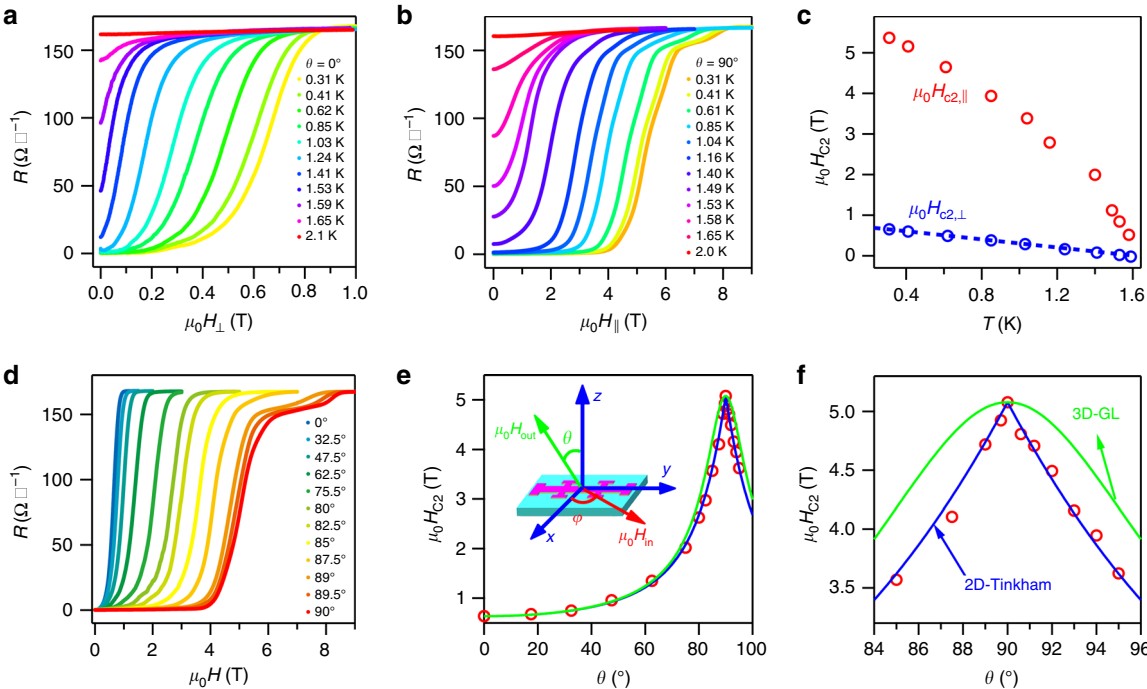

**Fig. 2** Two-dimensional superconductivity in few-layer $1T_d$-MoTe$_2$ crystals. **a, b** Superconducting resistive transition of the 8.6-nm-thick MoTe$_2$ crystal in perpendicular magnetic field (**a**) and in parallel magnetic field (**b**). **c** Temperature dependence of the upper critical field $\mu_0 H_{c2}$ corresponding to reduced resistance $r = 0.5$, with magnetic field directions parallel ($\mu_0 H_{c2,\parallel}$) and perpendicular ($\mu_0 H_{c2,\perp}$) to the crystal plane. The dashed line is fitting to the 2D Ginzburg–Landau theory. **d** Magnetic field dependence of the sheet resistance of the 8.6-nm MoTe$_2$ device at $T = 0.3$ K with different tilted angles $\theta$.

**e** Angular dependence of the upper critical field $\mu_0 H_{c2}$. The solid lines represent the fitting with the 2D Tinkham formula $\left| \frac{H_{c2}(\theta)\cos\theta}{H_{c2,\perp}} \right| + \left( \frac{H_{c2}(\theta)\sin\theta}{H_{c2,\parallel}} \right)^2 = 1$ (blue

line) and the 3D anisotropic mass model (3D-GL) $\left( \frac{H_{c2}(\theta)\cos\theta}{H_{c2,\perp}} \right)^2 + \left( \frac{H_{c2}(\theta)\sin\theta}{H_{c2,\parallel}} \right)^2 = 1$ (green line), respectively. The inset is a schematic drawing of the tilt

experiment setup, where $x$, $y$, and $z$ represents the crystallographic $b$-, $a$-, and $c$-axis, $\theta$ is the out-of-plane tilted angle between the out-of-plane magnetic field $B_{out}$ and the positive direction of $z$-axis, and $\varphi$ is the in-plane tilted angle between the in-plane magnetic field $B_{in}$ and the positive direction of $y$-axis.
**f** Zoom-in view of the region around $\theta = 90°$

temperature is estimated to be $T_{BKT} = 1.47$ K, which is only slightly larger than $T_{c,0} = 1.4$ K of the sample. With the above evidences, the 2D superconductivity is convincingly confirmed in our few-layer $1T_d$-MoTe$_2$ samples.

## Discussion

Now we turn to discuss the most important findings of our experiments, that is, the observation of the in-plane upper critical field $\left( H_{c2,\parallel} \right)$ beyond the Pauli limit and the emergent two-fold symmetry of $H_{c2,\parallel}$. In conventional BCS superconductors, sufficiently high external magnetic field can destroy the superconductivity by breaking Cooper pairs via the coexisting orbital[1,25] and Zeeman spin splitting effect[7,8]. For the few-layer sample, the orbital effect of the in-plane magnetic field is greatly suppressed due to the reduced dimensionality[1], and consequently $H_{c2,\parallel}$ is solely determined by the interaction between the external magnetic field and the spin of the electrons. When the magnetization energy gained from the applied magnetic field approaches to the superconducting condensation energy, the Cooper pairs are broken and superconductivity is destroyed at the characteristic field given by the Clogston–Chandrasekhar[7,8] or Pauli paramagnetic limit $H_P = \sqrt{2}\Delta_0/(g\mu_B)$, where $\Delta_0 = 1.76 k_B T_c$, $g$ is the $g$ factor, and $\mu_B$ as the Bohr magneton. The observation of the $H_{c2,\parallel}$ in our few-layer MoTe$_2$ is summarized in Fig. 3 for different samples. Figure 3a displays the superconducting transition in $1T_d$-MoTe$_2$ devices with various thicknesses under in-plane

magnetic field measured at 0.3 K. Clearly, the superconductivity in $1T_d$-MoTe$_2$ can persist to higher in-plane magnetic field as the thickness is lowered down. From Fig. 3b, c, it can be seen that the values of $H_{c2,\parallel}$ for the six typical samples with different thicknesses are all larger than $H_P$, in marked contrast to their bulk counterpart that is well below the $H_P$[20]. More generally, the magnetic field dependence of the sheet resistance of a 3-nm-thick MoTe$_2$ sample is further measured at $T = 0.3$ K ($T = 0.07 T_c$) with different in-plane tilted angle $\varphi$ as shown in Fig. 4a (see Supplementary Fig. 8 for $T = 0.3 T_c$, $0.6 T_c$, and $0.95 T_c$). Surprisingly, an emergent two-fold symmetry of $H_{c2,\parallel}$ has been observed in few-layer $1T_d$-MoTe$_2$ with the $H_{c2,\parallel}$ beyond $H_P$ in all the in-plane directions as shown in Fig. 4b. As the magnetic field tilted from $x$-axis ($\varphi = 0°$) to $y$-axis ($\varphi = 90°$) (the relation between the $x$- and $y$-axis and the crystal axis is shown in the Supplementary Fig. 9), we can see that the superconducting transition moves from higher field to lower field. From the low temperature ($T = 0.07 T_c$) to the temperature near $T_c$, the observed two-fold symmetry $H_{c2,\parallel}$ stays robust. The phenomena were observed in another two samples with the thickness of 4.0 and 9.0 nm, respectively (Supplementary Figs. 10 and 11). Note that the observed $H_{c2,\parallel}$ still stayed above the $H_P$, while the in-plane anisotropy, $H_{c2,\parallel}(0°)/H_{c2,\parallel}(90°)$, decreased from 1.56 to 1.16 with the thickness increased from 4 to 9 nm. For thicker samples, as superconductivity is also influenced by orbital effects, it is reasonable that the anisotropy in $H_{c2,\parallel}$ is reduced. The emergent two-fold symmetry observation in $H_{c2,\parallel}$, which exceeds $H_P$ at low

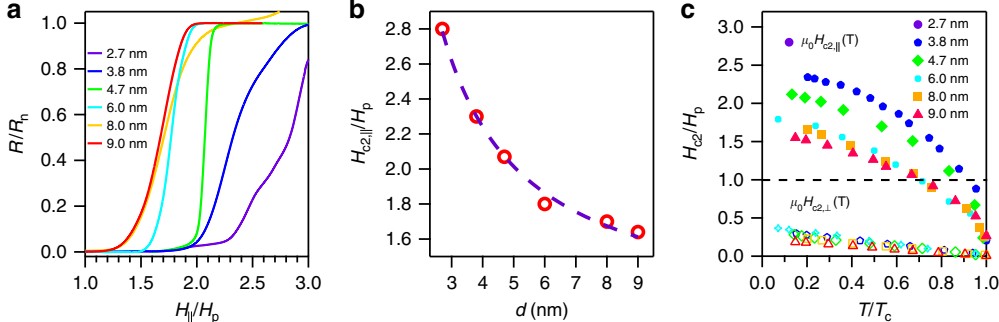

**Fig. 3** Enhanced in-plane upper critical field in few-layer $1T_d$-MoTe$_2$. **a** Magnetic field dependence of the resistance for $1T_d$-MoTe$_2$ devices with various thicknesses from 2.7 to 9 nm. The resistances and magnetic fields are normalized by the normal state resistance $R_n$ and the Pauli limit $H_p$, respectively. **b** Normalized in-plane upper critical field $H_{c2,\parallel}/H_p$ as a function of sample thickness $d$. The purple dashed line is a guide to the eye. **c** Normalized upper critical field $H_{c2}/H_p$ as a function of reduced $T/T_c$ for few-layer MoTe$_2$. The black dashed line denotes the Pauli limit $H_p$

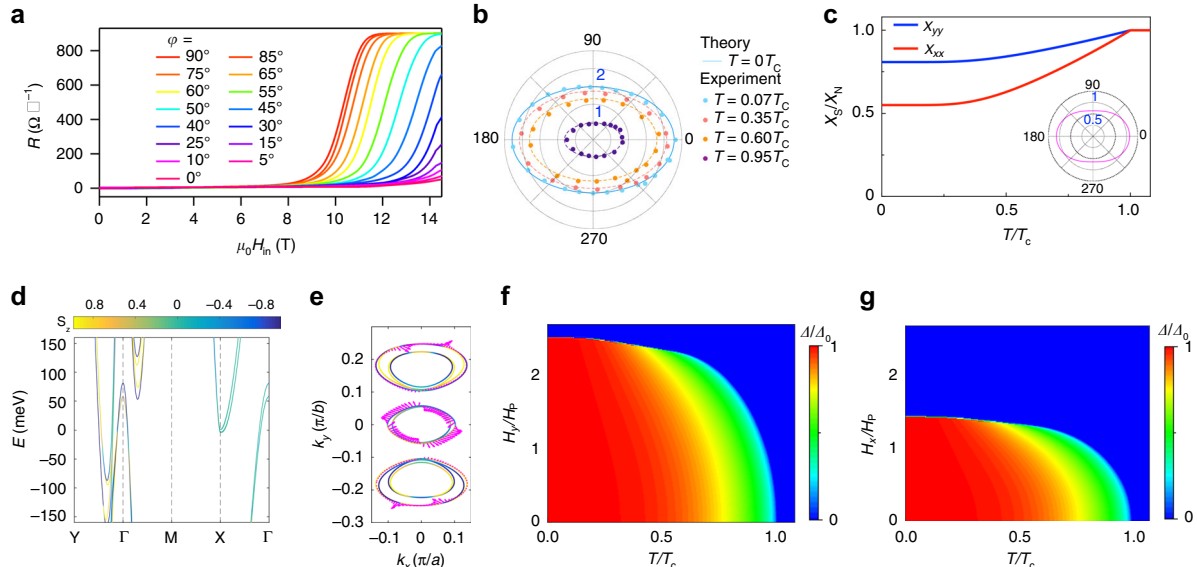

**Fig. 4** Two-fold symmetry of in-plane upper critical field $H_{c2,\parallel}$. **a** Magnetic field dependence of the sheet resistance of the 3-nm-thick MoTe$_2$ device at $T = 0.3\,\text{K}$ ($T = 0.07T_c$) with different in-plane tilted angles $\varphi$. **b** Angular dependence of the in-plane upper critical field normalized by Pauli limit $H_{c2,\parallel}/H_p$. The experimental data are measured at $0.07T_c$, $0.35T_c$, $0.6T_c$, and $0.95T_c$. The theoretical value of $H_{c2,\parallel}$ at $T = 0\,\text{K}$ is plotted to show the two-fold symmetry consistent with the experimental data at low temperature. The dashed lines are the asymptotic curves to show the two-fold symmetry maintains at $T = 0.35T_c$, $0.6T_c$, and $0.95T_c$. **c**, Temperature dependence of the normalized in-plane spin susceptibility $\chi_S/\chi_N$ along $x$ and $y$ direction, respectively. The inset is the polar plot for the zero temperature normalized spin susceptibility. **d** The first-principle calculations for the band structure of the bilayer $1T_d$-MoTe$_2$. The path $Y \to \Gamma \to M \to X \to \Gamma$ corresponds to the path $(0, 2\pi/b) \to (0, 0) \to (2\pi/a, 2\pi/b) \to (2\pi/a, 0) \to (0, 0)$ in the Brillouin zone, with $a$ and $b$ the lattice constant along $x$ and $y$ direction, respectively. The bands are labeled by out-of-plane spin polarization $<S_z>$. **e** The in-plane spin texture at the Fermi level. The in-plane spin–orbit coupling (SOC) is highly anisotropic at the $\Gamma$ pockets and the out-of-plane spin polarization dominates for the other two pockets. The color denotes the out-of-plane spin polarization $<S_z>$. **f, g** The temperature phase diagram for the superconducting state with anisotropic SOC under $y$- (**f**) and $x$- (**g**) oriented in-plane magnetic field, respectively

temperatures, is in sharp contrast with the standard BCS prediction and becomes highly nontrivial.

Similar anomalous enhancement of $H_{c2,\parallel}$ has been observed in layered superconductors in the dirty limit with strong SOC, which can be explained by spin–orbit scattering[26–31] (SOS) effect using the microscopic Klemm–Luther–Beasley (KLB) theory[30]. However, the isotropic SOS potential[30] in the KLB theory can only result in the isotropic $H_{c2,\parallel}$, which is inadequate to interpret our anisotropic $H_{c2,\parallel}$ data. The anisotropic SOS potential is one possibility that can lead to the observed anisotropy, but it requires the impurities to have the SOS potentials with a common two-

fold symmetry. This is unfeasible to realize in real materials. Moreover, the observed in-plane anisotropy is very robust and can be reproducible in many samples of different batches with different thickness, indicating that the effect is intrinsic rather than depending on some anisotropic disorder scattering. On the other hand, it is known that inhomogeneous superconducting states, such as Fulde–Ferrell–Larkin–Ovchinnikov (FFLO) state[32–35] or helical state[36], can also enhance $H_{c2,\parallel}$, which have been observed in heavy Fermion superconductors[32], organic superconductors[37,38], and monolayer Pb films[39]. However, for superconductors induced by FFLO state, the theoretical value[37] of

$H_{c2,\parallel}/H_p$ is in the range of 1.5~2.5, smaller than the observed value of 2.8 in our 2.7-nm-thick sample. Moreover, the FFLO characteristic upturn[32–34] of $H_{c2,\parallel}(T_c)$ at low temperature is missing in our experimental data as shown in Fig. 3c. Therefore, the FFLO state can be ruled out.

Recently in monolayer $NbSe_2$ and gated $MoS_2$ superconductor, the anomalous enhancement of $H_{c2,\parallel}$ beyond $H_p$ has been interpreted by the Ising SOC protected Ising superconductivity mechanism[2–4]. A monolayer TMDCs with $2H$ structure possesses an out-of-plane mirror symmetry, whereas the in-plane inversion symmetry is broken. The mirror symmetry restricts the crystal field ($\varepsilon$) to the plane, while the inversion symmetry breaking can induce strong SOC splitting, giving rise to an effective Zeeman-like magnetic field $H_{so}(k) \propto k \times \varepsilon$ (~100 T for gated $MoS_2$ and ~660 T for $NbSe_2$) with opposite out-of-plane direction at the K and –K valleys of the Brillouin zone[40]. Thus, the electron spins are pinned along the out-of-plane directions, and they are anti-parallel to each other for electrons with opposite momenta; that is to say, spins of electrons of Cooper pairs are polarized by the large out-of-plane effective Zeeman field, and thus becomes insensitive to the external in-plane magnetic field, which results in the enhancement of in-plane $H_{c2,\parallel}$. However, in few-layer $MoTe_2$, the absence of out-of-plane mirror symmetry gives rise to a more complicated SOC field beyond the Ising SOC.

In order to fully understand the experimental results in the few-layer $1T_d$-$MoTe_2$, we focus on the $1T_d$-$MoTe_2$ bilayer and construct an effective model from the symmetry point of view. The crystal structure of bilayer $1T_d$-$MoTe_2$ is shown in Supplementary Fig. 9 and it has the same symmetry properties as bulk crystals where only the mirror symmetry in the $y$ direction is preserved, while both the out-of-plane mirror symmetry and the in-plane mirror symmetry in the $x$ direction are broken. Since the bilayer $1T_d$-$MoTe_2$ respects the time reversal symmetry and the mirror symmetry in the $y$ direction (Supplementary Fig. 9), the SOC at the Fermi level is restricted to an effective form $H_{soc} = \mathbf{g} \cdot \boldsymbol{\sigma}$, with $\mathbf{g} = (x_1 \sin\varphi, y_1 \cos\varphi, z_1 \sin\varphi)$ in the first order approximation (the complete form is derived in the Supplementary Note 1), where $\varphi$ is the polar angle for the Fermi wave vector. In the bilayer $1T_d$-$MoTe_2$, the breaking of in-plane mirror symmetry in the $x$ direction generates the out-of-plane Ising component $g_z$, and the breaking of the out-of-plane mirror symmetry gives rise to the anisotropic in-plane components ($g_x$, $g_y$) of the SOC. As a result, all the three components of SOC exist in the bilayer $1T_d$-$MoTe_2$. Similar to $2H$ structure TMDCs, $g_z$ strongly enhances in-plane $H_{c2,\parallel}$. Interestingly, the in-plane anisotropic SOC will give rise to a two-fold symmetry in in-plane $H_{c2,\parallel}$ as discussed below.

To quantitatively validate the asymmetric SOC enhanced upper critical field, we calculate the in-plane spin susceptibility (see Supplementary Note 2 for the calculation procedure). At the zero temperature limit, the superconductor-–normal metal transition driven by in-plane magnetic field occurs when $\frac{1}{2}N_0\Delta_0^2 + \frac{1}{2}\chi_S H_\parallel^2 = \frac{1}{2}\chi_N H_\parallel^2$, where the two sides of the equation correspond to the energy for the superconducting state and the normal state, respectively. Here $\chi_S$ and $\chi_N$ denote the spin susceptibility of the superconducting state and the normal state, respectively, and $N_0$ is the density of states at the Fermi level. In the presence of the SOC, the superconducting spin susceptibility has a finite value as is seen from Fig. 4c. The enhancement of the in-plane $H_{c2}$ by the SOC field becomes understandable since $H_{c2}(\varphi) = \sqrt{\frac{N_0}{\chi_N - \chi_S}}\Delta_0$(Supplementary Note 3). The two-fold angle dependence of $H_{c2}$ is also consistently explained by the anisotropic in-plane spin susceptibility $\chi_S$ shown in the inset of Fig. 4c. In Fig. 4b, the in-plane $H_{c2}(\varphi)$ at zero temperature with two-fold symmetry is plotted with the SOC field at the Fermi level

$\mathbf{g} = k_B T_c (49 \sin\varphi, 68 \cos\varphi, 67 \sin\varphi)$ and fits well with the experimental data measured at the temperature $T = 0.07 T_c$. To substantiate the validity of the asymmetric SOC field, we further carry out the first-principle calculation for the bilayer $1T_d$-$MoTe_2$ and present the band structure as well as the anisotropic spin texture in Fig. 4d, e. In Fig. 4d, the spin bands splitting at the Fermi level agrees well with the estimated SOC field strength. In Fig. 4e, the in-plane spin texture consistently shows high anisotropy in all the Γ pockets and the other two pockets. The mean-field calculations for the pairing order parameter dependence on the in-plane magnetic field along $x$ and $y$ directions are further carried out to obtain the magnetic field–temperature phase diagram as shown in Fig. 4f, g. The $H_{c2}$ along $x$ and $y$ directions has strong anisotropy from zero temperature to near $T_c$ and shows the same trend as the experimental data measured in high temperature in Fig. 4b.

In summary, we demonstrate that the properties of the superconducting state of $1T_d$-$MoTe_2$ are strongly affected by a new type of asymmetric SOC which is in the order of tens of meV. Such strong SOC will create strong triplet pairing correlations in the material and may affect the pairing symmetry as well. Due to its large magnitude, the SOC may also have effects on the normal state spin transport of the system[41–43]. Importantly, the finding of the asymmetric SOC mostly depends on the symmetry of the crystal and similar asymmetric SOC are expected to exist in other $1T_d$ structure TMDCs such as the recently well-studied $1T_d$-$WTe_2$. Our findings on the new type of asymmetric SOC in $1T_d$-$MoTe_2$ are expected to promote further studies on the exotic superconducting and normal state phenomena in TMDCs, and boost the possible applications in superconducting spintronics[44,45,46] in TMDCs.

## Methods

**CVD synthesis of highly crystalline few-layer $MoTe_2$.** The few-layer $MoTe_2$ samples were synthesized via CVD method inside a furnace with a 1-in. diameter quartz tube. Specifically, one alumina boat containing precursor powder (NaCl: $MoO_3$ = 1:5) was put in the center of the tube. Si substrate with a 285-nm-thick $SiO_2$ on top was placed on the alumina boat with polished side faced down. Another alumina boat containing Te powder was put on the upstream side of quartz tube at a temperature of about 450 °C. Mixed gas of $H_2$/Ar with a flow rate of 15/80 sccm was used as the carrier gas. The furnace was ramped to 700 °C at a rate of 50 °C/min and held there for about 4 min to allow the growth of few-layer $MoTe_2$ crystals. After the reaction, the temperature was naturally cooled down to room temperature. All reagents were purchased from Alfa Aesar with purity exceeding 99%.

**Raman characterization.** Raman measurements with an excitation laser of 532 nm were performed using a WITEC alpha 300R Confocal Raman system. Before the characterization, the system was calibrated with the Raman peak of Si at 520 cm$^{-1}$. The laser power is <1 mW to avoid overheating of the samples.

**TEM and STEM characterization.** The STEM samples were prepared with a poly (methyl methacrylate) (PMMA)-assisted method. A layer of PMMA of about 1 μm thick was firstly spin coated on the wafer with $MoTe_2$ samples deposited, and then baked at 180 °C for 3 min. The wafer was then immersed in NaOH solution (1 M) overnight to etch the $SiO_2$ layer. After lift-off, the PMMA/$MoTe_2$ film was transferred into distilled (DI) water for several cycles to rinse off the residual contaminants, and then it was fished by a TEM grid (Quantifoil Au grid). The transferred specimen was dried naturally in ambient environment, and then dropped into acetone overnight to dissolve the PMMA coating layers. The STEM imaging on $MoTe_2$ were performed on a JEOL 2100F with a cold field-emission gun and a DELTA aberration corrector operating at 60 kV. A Gatan GIF Quantum was used to record the EELS spectra. The inner and outer collection angles for the STEM images (β1 and β2) were 62 and 129–140 mrad, respectively, with a convergence semi-angle of 35 mrad. The beam current was about 15 pA for the ADF imaging and EELS chemical analyses. All imaging was performed at room temperature.

**Devices fabrication and transport measurement.** Few-layer $MoTe_2$ samples were directly grown on $SiO_2$/Si substrate, which facilitate the device fabrication without the need for transferring the materials to an insulating substrate for transport measurement. After the growth of the sample, few-layer $MoTe_2$

crystals with the thickness ranging from 2 to 30 nm were firstly identified by their color contrast under optical microscopy. Then, small markers were fabricated using standard e-beam lithography near the identified sample for subsequent fabrication of Hall-bar devices. To obtain a clean interface between the electrodes and the sample, in situ argon plasma was employed to remove the resist residues before metal evaporation without breaking the vacuum. The Ti/Au (5/70 nm) electrodes were deposited using an electron-beam evaporator followed by lift-off in acetone. Transport experiments were carried out with a standard four-terminal method from room temperature to 0.3 K in a top-loading Helium-3 refrigerator with a 15 T superconducting magnet. A standard low-frequency lock-in technique was used to measure the resistance with an excitation current of 10 nA. Angular-dependent measurements were facilitated by an in situ home-made sample rotator. Due to the high reactivity of oxygen and water vapor, the few-layer $MoTe_2$ samples should be stored in an Ar-filled glove box to avoid deterioration once the device fabrication and transport measurement are finished.

## Data availability

The data that support the finding of this study are available from the corresponding author on request.

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

## Acknowledgements

We thank Xianxin Wu, Heng Fan, Jianlin Luo, Hsin Lin, and Jiangping Hu for stimulating discussions. This work has been supported by the National Basic Research Program of China from the MOST under the grant nos. 2016YFA0300600 and 2015CB921101, by the NSFC under the grant nos. 11527806 and 11874406. Research in Singapore was funded by the Singapore National Research Foundation under NRF award number NRF-RF2013-08, MOE Tier 2 MOE2016-T2-2-153, MOE2015-T2-2-007, and A*Star QTE program. J.L. and K.S. acknowledge JST-ACCEL and JSPS KAKENHI (JP16H06333 and P16382) for financial support. J.L. acknowledges financial support from the Hong Kong Research Grants Council (Project No. ECS26302118). K.L. thanks the support of the Croucher Foundation and HKRGC through grants C6026-16W, 16309718, 16307117, and 16324216. This research is partially supported by the Science, Technology, and Innovation Commission of Shenzhen Municipality (No. ZDSYS20170303165926217).

## Author contributions

J.C., P.L., J.Z., and W.-Y.H. contributed equally to this work. G.L. and Z.L. conceived and supervised the project, and designed the experiments; J.C., P.L., and X.H. fabricated the devices and carried out the transport measurements; J.Z. synthesized the sample; J.Y., J.F., Z.J., X.J., F.Q., and Z.G.C. made the transport measurement setup. J.L. and K.S. did the

measurements and data analysis on STEM; W.-Y.H. and K.T.L. predicted the presence of the anisotropic SOC and its experimental implications. J.L. performed the first-principle calculations. G.L. prepared the manuscript with input from Z.L., J.L., J.Z., W.-Y.H., K.T. L., J.L., C.Y., and L.L. All the authors discussed the results and commented on the manuscript.

## Additional information

**Competing interests:** The authors declare no competing interests.

