## [Peer Review File · Nature Communications]

Reviewers' comments:

Reviewer #1 (Remarks to the Author):

The authors study few-layer devices of 1Td-MoTe₂, a transition metal dichalcogenide (TMD) material synthesized using chemical vapor deposition. This manuscript is largely written as a material-science study, with detailed description on the synthesis, characterization, and thorough electronic transport measurements. Figures 1-3 confirm that this TMD is a well-behaved 2D superconductor, and the 2D properties agree with the now well understood Ising spin-orbit protection observed in ultrathin TMDs of the H type.

Although there is clear novelty in this work, I find that too little physical insight is gained to merit publication on Nature Communications.

To be specific, I find two point of novelty:

First, it is not obvious a-priori that the 1Td structure would give rise to the same protection seen in the H type structure. In this sense, this work demonstrates this nicely.

Second, and more importantly, is the in-plane anisotropy found in the upper critical field. Here, the reader is left unfulfilled: The actual novel dataset, presented in Figure 4, is supported by a plausible theoretical analysis based on symmetry. Although the theory agrees with experiment, there is very little which is done to further substantiate its validity. I would like to suggest several ideas, which the authors may use in a future submission:

1. In Figure S7, the authors measure the critical currents. Does the critical current exhibit the same anisotropy with respect to in-plane field? There is no need to fabricate devices with multiple probes here, but it would be interesting to look for changes in critical current with in-plane field angle.
2. It is not clear if the same effect is found at thicker samples. The authors should comment on this.
3. Can the authors rule out surface strain effects?

Finally, I believe some detail has to be presented about the device fabrication procedure. Since this is a CVD material, one assumes that this layer is transferred using the standard procedure, but the authors should clarify this.

Minor Comments:

"The understanding of this anisotropic SOC will help to understand the microscopic mechanism of nonlinear Hall effects recently observed in few-layer WTe₂." – it is not clear how the present study is related to these effects.

Figure 1d: It is not clear from the atomic model what is the difference between the 1T' and 1Td structures. I am probably missing something, but they look the same.

Figure 4c: Not clear is this is a theory plot or experiment.

Line 147: AMF → AFM

Line 180: "g as the g factor".

Reviewer #2 (Remarks to the Author):

The work by Cui et al. reports a transport study of superconducting 1Td-MoTe₂ in the 2D limit. They grow the samples by chemical vapor deposition, characterize them by Raman and STEM, and perform various transport measurements on nanofabricated Hall bars. They observe superconductivity with 2D character in few-layer devices and in-plane critical fields above the Pauli limit, indicating the relevance of spin-orbit coupling (SOC). In my opinion, the most important observation is that the critical field is different for fields applied in the x and y directions, indicating that the SOC is asymmetric. This is apparent in the bandstructure and the authors further substantiate it through their calculations of spin susceptibility.

Overall, I feel that there is a high degree of novelty in these experiments. The measurements seem carefully done and the data is both interesting and of high quality. I therefore strongly recommend publication in Nature Communications.

One minor comment is that I am confused by what $r = R/R_{5K}$ represents in the Table S1. What temperature is R measured?

Reviewer #3 (Remarks to the Author):

The major claim of this paper is summarized in Fig. 4. The authors present experimental evidence that the in-plane upper critical field $H_{c2||}$ of few-layer 1Td-MoTe₂ is anisotropic (i.e. that it is different, by roughly a factor of two, in the x- and y-directions). Moreover, they present theoretical evidence that in the *superconducting* state, there is an anisotropic spin susceptibility χ_s in the x- and y-directions whose origin is anisotropic spin-orbit coupling. They make the connection to the experimental data via $H_{c2||} \sim \sqrt{1/(\chi_N - \chi_s)}$. This is a novel observation, particularly in the context of the recent work on 2D TMD superconductors, and presents a new effect of the crystalline symmetry on the spin-orbit coupling in these materials. Spin-orbit coupling is essential in many interesting aspects of the TMD materials and so it is important to understand all of the ways in which it is manifest. I think this result will indeed influence thinking in this field. The arguments, based on fundamental crystalline symmetries (and a bilayer model), are rather convincing.

The paper is interesting and overall it is well done. With the inclusion of the theory, it is a vast improvement over the first version of their paper, which was on the arXiv some time ago. The paper is well organized and the figures make sense in the order in which they are presented. However, I have two suggestions about the organization of the paper that I think would improve the readability and accessibility:

- (a). Add a (slightly nicer) version of Fig. S10 to the main text (perhaps as part of Fig. 4). It would make the theory better integrated with the main paper, and it would go a long way to making it more accessible to physicists who are not expert in this field – seeing the spin polarization on the Fermi surface is a visceral way to understand the physics.
- (b). Move Fig. 2f and 2g to the supplementary. They don't really have much to do with the conclusions of the paper (I don't think anyone disputes that these behave as 2D superconductors).

Now, some points that the authors should address before the paper is published.

1. A fairly obvious question is: having shown the 1L, 2L and 3L samples in Figure 1, why do they do

no measurements on these? I assume it is because the few-layer samples degrade rather quickly in ambient conditions, but the authors should comment on this.

2. The authors appear to have discounted the possibility of spin-orbit scattering (SOS) based on the fact that the square resistance of their sample is much less than $h/4e^2$ approaching the superconducting transition. However, the Hall measurement tells a different story: they deduce a mean free path of about 10 nm, which is smaller than any coherence length they measure (Table S1). It appears from this that they are not in the limit of a clean superconductor. The authors should comment on this, and how this relates to their conclusion that SOS plays no role in their $H_{c2||}$ result.

3. Fig. 3b is rather interesting, but it leads one to the rather strange conclusion that in the very thick limit, the $H_{c2||}$ might be larger than H_p . Even the “guide to the eye” they have drawn is continued with a straight line, this seems to extrapolate to $d = 25$ nm or so. The authors have a good opportunity here to study the crossover from 2D to 3D superconductivity, and they do indeed have a 30 nm sample. Do they have any $H_{c2||}$ data from thicker samples? Is there any evidence that there is enhancement of $H_{c2||}$ in the thicker samples? The interesting spin texture seems to suggest that there might be some $H_{c2||}$ enhancement even in the 3D limit. The authors should comment on this – it is an obvious omission from their work.

Finally, some specific comments:

- Fig. 2c, caption: the authors should note what angle ϕ was used to determine the $H_{c2||}$.

- Fig. S11 is a bit sloppy. It would be good to have some more axis labels and to indicate where the authors are getting their numbers for the SOC splitting (perhaps by labelling the bands by their spin projections).

We thank the editor and referees for the constructive comments on our manuscript. We have addressed all the comments point-by-point and revised the manuscript accordingly. In this response letter, comments from the referees are in black typeface, and our responses are in blue typeface. All major changes have been highlighted in red in the main text and Supplemental Information (SI).

New experiments include in-plane anisotropic critical current investigation, and the effect of sample thickness and surface strain on the in-plane anisotropy, both of which have been added in the manuscript to further support our claim. **Two new figures** have been added into the revised SI.

Response to the referees:

Referee #1

The authors study few-layer devices of 1Td-MoTe₂, a transition metal dichalcogenide (TMD) material synthesized using chemical vapor deposition. This manuscript is largely written as a material-science study, with detailed description on the synthesis, characterization, and thorough electronic transport measurements. Figures 1-3 confirm that this TMD is a well-behaved 2D superconductor, and the 2D properties agree with the now well understood Ising spin-orbit protection observed in ultrathin TMDs of the H type.

Although there is clear novelty in this work, I find that too little physical insight is gained to merit publication on Nature Communications.

Response: We thank the reviewer for the constructive comments and suggestions. It helps us a lot to improve the quality of our manuscript. According to the suggestion, we have carried out the critical current measurements with respect to in-plane field to demonstrate the same anisotropy as the one shown in upper critical fields. Accordingly, we have revised our manuscript by re-organizing the text structure and adding physical insight to emphasize the importance and novelty of our work.

Comment #1:

To be specific, I find two point of novelty:

First, it is not obvious a-priori that the 1Td structure would give rise to the same protection seen in the H type structure. In this sense, this work demonstrates this nicely.

Response #1:

We thank the reviewer for the positive comments. As the referee pointed out, it is not obvious that 1Td structures would give rise to the enhancement of $H_{c2,\parallel}$. Here we experimentally and theoretically demonstrated that the enhancement of $H_{c2,\parallel}$ is due to a new type of anisotropic spin-orbit coupling.

Comment #2:

Second, and more importantly, is the in-plane anisotropy found in the upper critical field. Here, the reader is left unfulfilled: The actual novel dataset, presented in Figure 4, is supported by a plausible theoretical analysis based on symmetry. Although the theory agrees with experiment, there is very little which is done to further substantiate its validity.

Response #2:

We thank the reviewer for the comments. We agree with the referee that the presence of anisotropic SOC in 1Td structure is a key result of our work and the claim should be substantiated. Indeed, we have done several things to support our claim:

a) We obtained the asymmetric SOC from symmetry arguments which is very general. With the asymmetric SOC, two key experimental findings are explained. First, the enhancement of the in-plane $H_{c2,\parallel}$. Second, the asymmetric in-plane $H_{c2,\parallel}$ (indeed, the asymmetric $H_{c2,\parallel}$ was first predicted theoretically and then experimentally verified).

b) The asymmetric spin-orbit coupling (SOC) is further confirmed by our first-principle band structure calculation as showed in the SI in the previous version (in the new version, the first principle calculation results are shown in the main text). Importantly, at the Fermi energy, our first-principle calculation on the spin bands splitting gives similar values as our estimation of the strength of the spin-orbit coupling (SOC). Moreover, the asymmetric spin texture from symmetry analysis is confirmed by the first-principle calculation, which further substantiates the validity of the symmetry analysis.

c) With the asymmetric spin-orbit coupling (SOC), we performed mean-field calculations and predicted that it leads to a two-fold symmetry for the in-plane $H_{c2,\parallel}$ which was also verified experimentally.

In the revised main text, we re-organized the text and presented the asymmetric spin texture as well as the spin resolved band structure from the first-principle calculation in Fig. 4d and 4e and emphasized the consistency of our symmetry analysis and the first-principle calculation. In the

revised SI, we added the mean field calculation for the pairing order parameter as the function of in-plane magnetic field.

Comment #3:

I would like to suggest several ideas, which the authors may use in a future submission:

In Figure S7, the authors measure the critical currents. Does the critical current exhibit the same anisotropy with respect to in-plane field? There is no need to fabricate devices with multiple probes here, but it would be interesting to look for changes in critical current with in-plane field angle.

Response #3:

We appreciate the valuable comment and suggestion which improve the coherence of our manuscript. As suggested, we **fabricated a fresh 4-nm-thick MoTe₂ device (inset of Fig. R1a)**. Fig. R1a and R1b show the measurement results of the in-plane upper critical field $H_{c2,\parallel}(\varphi)$ under different in-plane tilted angles φ . It can be seen that $H_{c2,\parallel}(\varphi)$ shows a clear two-fold symmetry with an anisotropy represented by $H_{c2,\parallel}(0^\circ)/H_{c2,\parallel}(90^\circ)=1.56$, which is consistent with our previous report. **Fig. R1c shows the in-plane critical current $I_{c,\parallel}(\varphi)$ of the same device under an in-plane field $B_{\parallel}=1$ T, 3 T, and 5 T.** As expected, the in-plane critical current $I_{c,\parallel}(\varphi)$ is notably suppressed by the application of external magnetic field for a certain tilted angle. At each field, $I_{c,\parallel}(\varphi)$ increases from a minimum to a maximum as the in-plane field tilted angle φ changes from 90° to 0° . To make it clear, we plot the in-plane critical current $I_{c,\parallel}(\varphi)$ as a function of in-plane tilted angle φ in polar coordinates in Fig. R1d. **As in the case of the in-plane upper critical field $H_{c2,\parallel}$, the in-plane critical current also exhibits two-fold symmetry,** indicating that an anisotropic in-plane critical current existed in few-layer 1Td-MoTe₂. More

importantly, we find that the anisotropy of in-plane critical current represented by $I_{c,\parallel}(0^\circ)/I_{c,\parallel}(90^\circ)$ increases from 1.25 to 3.0 as the in-plane magnetic field increases from 1 T to 5 T (Fig. R1d). This indicates that the superconducting gap is suppressed by the in-plane field and such suppression is crystal directions dependent. **We propose that the origin of the two-fold symmetry observed in in-plane critical current and upper critical field is a result of asymmetric spin-orbit coupling (SOC).** The figure and discussion are added in SI as Figure S10.

Fig. R1 (a) Magnetic field dependence of the longitudinal resistance of the 4-nm-thick MoTe₂ device measured at $T = 0.3$ K ($T = 0.2 T_c$) under typical in-plane tilted angles ϕ . Inset: Optical image of the 4-nm-thick MoTe₂ device. (b) Angular dependence of the in-plane upper critical field normalized by Pauli limit $H_{c2,\parallel}/H_P$. (c) Tilted angle ϕ dependence of the in-plane critical

current $I_{c,\parallel}(\varphi)$ measured at $T = 0.3$ K under in-plane field of 1 T, 3 T, and 5 T. (d) Angular dependence of the in-plane critical current $I_{c,\parallel}(\varphi)$ under in-plane field of 1 T, 3 T, and 5 T.

Comment #4:

It is not clear if the same effect is found at thicker samples. The authors should comment on this.

Response #4:

From Fig. R1, the in-plane anisotropy of the upper critical field represented by $H_{c2,\parallel}(0^\circ)/H_{c2,\parallel}(90^\circ)$ is found to be 1.56 for 4.0-nm-thick sample, which agrees well with the in-plane anisotropy of 1.53 observed in 3.0-nm-thick sample (Figure 4b in the revised manuscript). Thus, **the in-plane anisotropy can be easily observed in relatively thin samples ($d < 5$ nm).**

To study the influence of thickness on the in-plane anisotropy, we fabricated another device with the thickness of ~ 9 nm. The measured magnetoresistance R and extracted in-plane upper critical field $H_{c2,\parallel}(\varphi)$ with respect to in-plane tilted angles φ are shown in **Fig. R2**. Though the two-fold symmetry can be observed, its amplitude is found to decrease sharply from 1.56 to 1.16, indicating that **the in-plane anisotropy strongly depends on the sample thickness**. In the 9-nm-thick sample, since the $H_{c2,\parallel}$ is still much higher than the Pauli limit H_p , it indicates that SOC still plays an important role. However, superconductivity is also influenced by orbital effects in the thicker samples, it's reasonable that the anisotropy in $H_{c2,\parallel}$ is reduced. Figure S11 and discussion are added into the SI to highlight the influence of sample thickness on the in-plane anisotropy.

Fig. R2 (a) Magnetic field dependence of the sheet resistance R of the 9-nm-thick MoTe_2 device at $T = 0.3 \text{ K}$ ($T = 0.07 T_c$) with typical in-plane tilted angles ϕ . (b) Optical image of the 9-nm-thick MoTe_2 device. The red rectangular indicates the measured sample. (c) Angular dependence of the in-plane upper critical field normalized by Pauli limit $H_{c2,\parallel}/H_P$.

Comment #5:

Can the authors rule out surface strain effects?

Response #5:

It is known that strain may affect the vibrational modes³⁻⁶. Therefore, recently Raman spectra were successfully employed to study the strain in two-dimensional materials⁷⁻¹⁰ such as graphene and MoS_2 . Theoretically, Johnson *et al*¹¹ have studied the influence of SiO_2 substrate on monolayer MoTe_2 based on first-principles calculations and found a negligible strain effect. The Raman experiments performed on different substrates including SiO_2 , Quartz, and Sapphire lead

to the same conclusion that the substrates have negligible effect on few-layer MoTe₂¹². From Fig. 1b in the revised manuscript, we can find that Ag modes at 127, 161 cm⁻¹ shows no discernable shifts with increasing layer number. The small shift of Ag mode at 267 cm⁻¹ comes from the interlayer pairing^{10,13} rather than the surface strain effect. However, high-pressure Raman experiments¹⁴ signify that pressure-induced strain can dramatically modify the Ag modes. Therefore, we can conclude that the surface strain plays a negligible effect on our few-layer 1Td-MoTe₂.

Moreover, as explained in **the reply to Comment #4**, the anisotropy is strongly thickness dependent. This also suggests that the observed anisotropy should not come from the surfaces but most likely from the bulk.

Comment #6:

Finally, I believe some detail has to be presented about the device fabrication procedure. Since this is a CVD material, one assumes that this layer is transferred using the standard procedure, but the authors should clarify this.

Response #6:

We thank the reviewer for careful and thorough reading of this manuscript. Different from other CVD materials grown on metal substrate (such as graphene grown on Cu foil), **our few-layer MoTe₂ samples are directly deposited on SiO₂/Si substrate**, which enables us to directly fabricate the device right after the growth of the sample. Therefore, classic transfer is not required in our work.

Minor Comments:

Minor Comment #1: “The understanding of this anisotropic SOC will help to understand the microscopic mechanism of nonlinear Hall effects recently observed in few-layer WTe₂.” – it is not clear how the present study is related to these effects.

Response: Thanks for the comment. In the revised version, we removed this statement.

Minor Comment #2: Figure 1d: It is not clear from the atomic model what is the difference between the 1T' and 1Td structures. I am probably missing something, but they look the same.

Response: The unit cell of 1T' phase is monoclinic which has global inversion center, while the 1Td phase is orthorhombic which lacks inversion symmetry. So, from symmetry point of view, the two models describing the 1T' and the 1Td phases are very different.

It can be considered that there is a slightly translational shift in each layer between 1T' and 1Td phases in the unit cell structure. For monolayer MoTe₂, 1T' is the same as 1Td phase since the stacking can be omitted.

Minor Comment #3: Figure 4c: Not clear is this is a theory plot or experiment.

Response: Thanks for the comment. The Fig. 4c is the theoretical calculated in-plane spin susceptibility.

Minor Comment #4: Line 147: AMF → AFM

Response: We already corrected the typo.

Minor Comment #5: Line 180: “g as the g factor”.

Response: We have corrected the typo.

Referee #2

The work by Cui et al. reports a transport study of superconducting 1Td-MoTe₂ in the 2D limit. They grow the samples by chemical vapor deposition, characterize them by Raman and STEM, and perform various transport measurements on nanofabricated Hall bars. They observe superconductivity with 2D character in few-layer devices and in-plane critical fields above the Pauli limit, indicating the relevance of spin-orbit coupling (SOC). In my opinion, the most important observation is that the critical field is different for fields applied in the x and y directions, indicating that the SOC is asymmetric. This is apparent in the band structure and the authors further substantiate it through their calculations of spin susceptibility.

Overall, I feel that there is a high degree of novelty in these experiments. The measurements seem carefully done and the data is both interesting and of high quality. I therefore strongly recommend publication in Nature Communications.

Response: We appreciate your positive comments.

Comment #1:

One minor comment is that I am confused by what $r = R/R_{5K}$ represents in the Table S1. What temperature is R measured?

Response #1:

Sorry for the confusion. We introduce the reduced resistance of $r = R/R_N = R/R_{5K}$ to facilitate the definition of the critical transition temperatures $T_{c,r}$ for different superconducting fractions (see the main text in revised manuscript). R_{5K} is the resistance measured at $T=5$ K and R varies as a function of temperature.

Referee #3

The major claim of this paper is summarized in Fig. 4. The authors present experimental evidence that the in-plane upper critical field $H_{c2,\parallel}$ of few-layer 1Td-MoTe₂ is anisotropic (i.e. that it is different, by roughly a factor of two, in the x - and y -directions). Moreover, they present theoretical evidence that in the superconducting state, there is an anisotropic spin susceptibility χ_s in the x - and y -directions whose origin is anisotropic spin-orbit coupling. They make the connection to the experimental data via $H_{c2,\parallel} \sim \sqrt{\frac{1}{\chi_N - \chi_s}}$. This is a novel observation, particularly in the context of the recent work on 2D TMD superconductors, and presents a new effect of the crystalline symmetry on the spin-orbit coupling in these materials. Spin-orbit coupling is essential in many interesting aspects of the TMD materials and so it is important to understand all of the ways in which it is manifest. I think this result will indeed influence thinking in this field.

The arguments, based on fundamental crystalline symmetries (and a bilayer model), are rather convincing.

The paper is interesting and overall it is well done. With the inclusion of the theory, it is a vast improvement over the first version of their paper, which was on the arXiv some time ago. The paper is well organized and the figures make sense in the order in which they are presented.

Response: We appreciate the reviewer for the positive comments and constructive suggestions.

Comment #1:

However, I have two suggestions about the organization of the paper that I think would improve the readability and accessibility:

(a). Add a (slightly nicer) version of Fig. S10 to the main text (perhaps as part of Fig. 4). It would make the theory better integrated with the main paper, and it would go a long way to making it more accessible to physicists who are not expert in this field – seeing the spin polarization on the Fermi surface is a visceral way to understand the physics.

(b). Move Fig. 2f and 2g to the supplementary. They don't really have much to do with the conclusions of the paper (I don't think anyone disputes that these behave as 2D superconductors).

Response #1: We thank the reviewer for these constructive suggestions. It really helps a lot to improve readability and accessibility of our manuscript. Following the reviewer's suggestions, we have moved Supplementary Figure S10 and S11 to the main text, and incorporate them into Fig. 4 in the revised version. For consistency, Figure 2f and 2g have been moved to the SI. Accordingly, we re-organized the text structure in revised manuscript.

Comment #2:

Now, some points that the authors should address before the paper is published.

1. A fairly obvious question is: having shown the 1L, 2L and 3L samples in Figure 1, why do they do no measurements on these? I assume it is because the few-layer samples degrade rather quickly in ambient conditions, but the authors should comment on this.

Response #2: We thank the reviewer's careful reading and comment. As the reviewer pointed out, due to the high reactivity of oxygen and water vapor, the few-layer MoTe_2 samples degrade quickly in ambient conditions. Therefore, the devices with 1L~2L thickness are not electrically conducting due to sample deterioration. In the revised version, we added comments on the stability of the measured sample in Materials and Methods section.

Comment #3:

2. The authors appear to have discounted the possibility of spin-orbit scattering (SOS) based on the fact that the square resistance of their sample is much less than $h/4e^2$ approaching the superconducting transition. However, the Hall measurement tells a different story: they deduce a mean free path of about 10 nm, which is smaller than any coherence length they measure (Table S1). It appears from this that they are not in the limit of a clean superconductor. The authors should comment on this, and how this relates to their conclusion that SOS plays no role in their $H_{c2,\parallel}$ result.

Response #3: Thanks for the comment. We agree that our samples are not in the clean limit due to the mean free path is smaller than the coherence length, but our anisotropic $H_{c2,\parallel}$ data cannot be fully interpreted by the SOS mechanism in the KLB¹⁵ theory. In the KLB theory¹⁵, the spin-orbit scattering (SOS) randomizes the spin direction and weakens the Pauli paramagnetism, so it provides the possibility to have an in-plane H_{c2} beyond the Pauli limit. However, **the isotropic SOS potential¹⁵ in the KLB theory can only result in the isotropic $H_{c2,\parallel}$** , which is inadequate to interpret our anisotropic $H_{c2,\parallel}$ data. The anisotropic SOS potential is one possibility that can lead to the observed anisotropy, but it requires the impurities to have the SOS potentials with a common two-fold symmetry. This is unfeasible to realize in real materials. The observed in-plane anisotropy is very robust and can be reproducible in samples of different batches with different thickness. As a result, we consider the intrinsic anisotropic spin-orbit coupling from MoTe₂ as the direct reason for our anisotropic $H_{c2,\parallel}$ result. In the revised manuscript, we emphasize the fact that our anisotropic $H_{c2,\parallel}$ result is unrealistic to be explained by the SOS mechanism. We have deleted the notion about our samples are all in the low-disorder regime in the revised version.

Comment #4:

3. Fig. 3b is rather interesting, but it leads one to the rather strange conclusion that in the very thick limit, the $H_{c2,\parallel}$ might be larger than H_p . Even the “guide to the eye” they have drawn is continued with a straight line, this seems to extrapolate to $d = 25$ nm or so. The authors have a good opportunity here to study the crossover from 2D to 3D superconductivity, and they do indeed have a 30 nm sample. Do they have any $H_{c2,\parallel}$ data from thicker samples? Is there any evidence that there is enhancement of $H_{c2,\parallel}$ in the thicker samples? The interesting spin texture seems to suggest that there might be some $H_{c2,\parallel}$ enhancement even in the 3D limit. The authors should comment on this – it is an obvious omission from their work.

Response #4:

The referee is right that the $H_{c2,\parallel}$ can be higher than the Pauli limit H_p even for thicker samples. This requires the out-of-plane superconducting coherence length to be short enough so that the orbital depairing upper critical field can be larger than the Pauli limit. Then the orbital depairing becomes less detrimental and the real upper critical field is determined by the paramagnetic field. In this case, the SOC can enhance the upper critical field even for thick samples. This is the case of many layered (bulk) superconductors which have $H_{c2,\parallel}$ larger than the Pauli limit, like TaS₂-Py, TaS₁Se₁, NbSe₂, shown in Fig. 4F in Science 350, 1535 (2015). Since the key point of the present manuscript is to study the asymmetric SOC effect on the superconducting properties, we focus our studies on thinner samples to suppress the orbital effects. A systematic study of the superconducting properties on the thickness dependence to explore the 2D to 3D crossover is an interesting research direction but it is beyond the scope of our current manuscript. We thank the referee for pointing out the interesting research direction.

Comment #5:

Finally, some specific comments:

Fig. 2c, caption: the authors should note what angle ϕ was used to determine the $H_{c2,\parallel}$.

Response #5: We thank the reviewer's comment. The out-of-plane tilted experiment was carried out at arbitrary angle ϕ , and it does not affect the conclusion of two-dimensional superconductivity. For convenience, we introduced the angle ϕ in advance (inset of Fig. 2e in revised manuscript) to study the two-fold symmetry of in-plane anisotropy.

Comment #6:

Fig. S11 is a bit sloppy. It would be good to have some more axis labels and to indicate where the authors are getting their numbers for the SOC splitting (perhaps by labelling the bands by their spin projections).

Response #6: We thank the reviewer's comment. We move the original Supplementary Figure S10 and S11 to Fig. 4 as new Fig. 4d and 4e in the main text. The bands are all labeled by the spin projection along the z -axis direction and the SOC splitting at the Fermi level can be accordingly read out. We get the numbers for the SOC field by fitting it to the experimentally measured in-plane $H_{c2,\parallel}$ through angle-dependent spin susceptibility $\chi_{\phi\phi}^S$ formula and $H_{c2,\parallel}(\phi)$ formula in the SI. We add one sentence in the revised Supplementary Information to illustrate how we get the optimized numbers for the SOC field. The estimated SOC field strength matches well with the SOC induced bands splitting at the Fermi level from the first-principle calculation in Fig. 4d.

References:

1. Xi, X. X. *et al.* Ising pairing in superconducting NbSe₂ atomic layers. *Nat. Phys.* **9**, 139-143 (2016).
2. Barrera, S. C. *et al.* Tuning Ising superconductivity with layer and spin-orbit coupling in two-dimensional transitionmetal dichalcogenides. *Nat. Commun.* **9**, 1427 (2018).
3. Mohiuddin, T. M. G. *et al.* Uniaxial strain in graphene by Raman spectroscopy: G peak splitting, Grüneisen parameters, and sample orientation. *Phy. Rev. B* **79**, 205433 (2009).
4. Castellanos-Gomez, A. *et al.* Local Strain Engineering in Atomically Thin MoS₂. *Nano Lett.* **13**, 5361-5366 (2013).
5. Ni, Z. H. *et al.* Uniaxial Strain on Graphene: Raman Spectroscopy Study and Band-Gap Opening. *ACS Nano* **2**, 2301-2305 (2008).
6. Huang, M. Y. *et al.* Phonon softening and crystallographic orientation of strained graphene studied by Raman spectroscopy. *Proc. Natl. Acad. Sci. U.S.A.* **106**, 7304-7308 (2009).
7. Robinson, J. A. *et al.* Raman Topography and Strain Uniformity of Large-Area Epitaxial Graphene. *Nano Lett.* **9**, 964-968 (2009).
8. Ferralis, N. *et al.* Evidence of Structural Strain in Epitaxial Graphene Layers on 6H-SiC(0001). *Phy. Rev. Lett.* **101**, 156801 (2008).
9. Yu, T. *et al.* Raman Mapping Investigation of Graphene on Transparent Flexible Substrate: The Strain Effect. *J. Phys. Chem. C* **112**, 12602 (2008).
10. Li, H. *et al.* From Bulk to Monolayer MoS₂: Evolution of Raman Scattering. *Adv. Mat.* **22**, 1385-1390 (2012).
11. Naylor, C. H. *et al.* Monolayer Single-Crystal 1T'-MoTe₂ Grown by Chemical Vapor Deposition Exhibits Weak Antilocalization Effect. *Nano Lett.* **16**, 4297-4304 (2016).
12. Song, Q. J. *et al.* Physical origin of Davydov splitting and resonant Raman spectroscopy of Davydov components in multilayer MoTe₂. *Phy. Rev. B* **93**, 115409 (2016).
13. Luo, X. *et al.* Anomalous frequency trends in MoS₂ thin films attributed to surface effects. *Phy. Rev. B* **88**, 075320 (2013).
14. Qi, Y. P. *et al.* Superconductivity in Weyl semimetal candidate MoTe₂. *Nat. Commun.* **7**, 11038 (2015).
15. Klemm, R. A. *et al.* Theory of upper critical field in layered superconductors. *Phys. Rev. B* **12**, 877-891 (1975).

REVIEWERS' COMMENTS:

Reviewer #1 (Remarks to the Author):

I find that the authors have fully addressed my concern from the previous round.
This manuscript can be published in Nature Communications.

Reviewer #3 (Remarks to the Author):

I am satisfied that the authors have addressed my comments and I recommend publication in Nat Comm.

One comment before the proof stage of the paper: please take more care with the axis and legend labels in all the figures, but especially Fig. 4. I can barely read them.

We thank the editor and referees for the constructive comments on our manuscript. We have addressed all the issues point-by-point and revised the manuscript accordingly. In this response letter, comments from the referees are in black typeface, and our responses are in blue typeface.

Response to the referees:

Referee #1

I find that the authors have fully addressed my concern from the previous round.

This manuscript can be published in Nature Communications.

Response: We thank the reviewer for supporting publication in Nature Communications.

Referee #3

I am satisfied that the authors have addressed my comments and I recommend publication in Nature Communications.

Response: We appreciate your recommendation to publish in Nature Communications.

Comment #1:

One comment before the proof stage of the paper: please take more care with the axis and legend labels in all the figures, but especially Fig. 4. I can barely read them.

Response #1: According to the suggestion of the referee, we have enlarged the legend labels in all the figures.